**Funding:** The author(s) received no specific funding for this work.

**Competing interests:** Financial support was provided by the Fundação de Amparo à Pesquisa

# Measurement properties of outcome measures used in neurological telerehabilitation: A systematic review protocol

**Sherindan Ayessa Ferreira de Brito**[☯]**, Aline Alvim Scianni**[☯]**, Paula da Cruz Peniche**[☯]**, Christina Danielli Coelho de Morais Faria**[ID]*[☯]

Department of Physical Therapy, Universidade Federal de Minas Gerais, Belo Horizonte, Minas Gerais, Brazil

☯ These authors contributed equally to this work.
* cdcmf@ufmg.br

## Abstract

Several measurement tools commonly used in face-to-face neurological rehabilitation have been used in telerehabilitation. However, it is not known whether these tools have adequate measurement properties and clinical utility. This systematic review aims to investigate the measurement properties and the clinical utility of measurement tools used in telerehabilitation in individuals with neurological diseases. A systematic review to investigate the measurement properties and clinical utility of measurement tools used in telerehabilitation in individuals with neurological conditions will be conducted. This systematic review will follow the Preferred Reporting Items for Systematic Review and Meta-Analyses (PRISMA) statement. this systematic review protocol was registered in the International Prospective Register of Systematic Reviews (PROSPERO) on 28 May 2021 (registration number: CRD42021257662). Electronic searches will be performed in following databases: Medical Literature Analysis and Retrieval System Online (MEDLINE Ovid), Excerpta Medica Database (Embase Classic + Embase Ovid), Physiotherapy Evidence Database (PEDro), Scientific Electronic Library Online (Scielo), and Literatura Latino-Americana e do Caribe em Ciências da Saúde (LILACS). Two trained independent reviewers will select the studies according to the inclusion criteria, and will also extract the data, evaluate the clinical utility and methodological quality. The relevant data such as design, participants, settings, and mode of administration, measurement properties, and clinical utility will be summarized. Disagreements between reviewers will be resolved by consensus or by the decision of a third independent reviewer. Hand searches of other relevant studies will be employed. The COnsensus-based Standards for the selection of health Measurement Instruments (COSMIN) checklist and the clinical utility scale will be used to assess the methodological quality and clinical utility of these tools, respectively. This systematic review will provide information regarding the measurement properties and the clinical utility of the measurement tools used in neurological telerehabilitation. This information will be useful to assist health professionals in choosing adequate measurement tools and planning new research studies.

do Estado de Minas Gerais (FAPEMIG), Coordenação de Aperfeiçoamento de Pessoal de Nível Superior (CAPES – Finance Code 001), Conselho Nacional de Desenvolvimento Científico e Tecnológico (CNPQ) and Pró-reitoria de Pesquisa da Universidade Federal de Minas Gerais (PRPq/UFMG).

## Introduction

Neurological conditions have a high prevalence and incidence [1]. These disorders are the leading cause of disabilities and the second biggest cause of death worldwide [1]. From 1990 to 2016, the disabilities and deaths caused by neurological conditions increased approximately 39% and 15%, respectively [1]. It leads to impairments, activity limitations, and participation restrictions [2], as well as negative impacts on the quality of life [3–6]. This makes these individuals commonly in need of rehabilitation services. According to Cieza et al. 2021, neurological conditions were one of the largest determinants of the need for rehabilitation [7]. However, they usually face barriers to access the rehabilitation centers, such as transport problems, lack of a caregiver, high costs, among others [8–10]. In contexts as the COVID-19 pandemic, these barriers may be even greater, as social distancing is recommended to avoid the spread of the virus [11]. Therefore, telerehabilitation has been an innovative alternative to delivery rehabilitation services.

Telerehabilitation can be defined as the remote delivery of rehabilitation services using information and communication technologies, such as telephone, videoconferencing, and sensors [12]. Telerehabilitation sessions can be used to assess, goal-setting, intervention, education, and monitoring [2]. The use of telerehabilitation has increased over time as technologies become increasingly prevalent and easily accessible [12].

Telerehabilitation has several benefits, such as easier access, can be carried out at the individual's home, does not require transportation [13], and has a lower cost when compared to face-to-face rehabilitation [14–16]. In recent years, the use of telerehabilitation has increased, mainly due to the development of new computer technologies and more advanced devices that allow long-distance communication [12, 16]. Moreover, during the COVID-19 pandemic, this strategy was even more widespread.

Recent systematic reviews assessed the effectiveness of telerehabilitation in individuals with neurological conditions [17–19]. In these reviews, various outcomes were investigated, and many measurement tools commonly applied in the face-to-face evaluation were used, such as Barthel Index, Berg Balance Scale, Fugl-Meyer Upper Extremity, Action Research Arm Test, and Stroke Impact Scale [17–19]. However, assessing remotely an individual can be challenging [20].

It is important to consider that the results of these measurement tools may be different when comparing telerehabilitation and face-to-face rehabilitation. The measurement tools commonly used in face-to-face rehabilitation may require adaptations for use in telerehabilitation [20]. For example, the Fulg-Meyer Scale [21] commonly used in the face-to-face evaluation of individuals after stroke can be a challenge to remote use [20]. In addition, communication, comprehension, and interaction are different remotely when compared to face-to-face [22]. All these factors can interfere with the measurements provided by the tools. Hence, it is important to establish the measurement properties of the measurement tools, when used in telerehabilitation. The choice of tools with adequate and accurate measurement properties is important to guarantee the quality and reliability of the results, both in research and in clinical practice [23].

As previously discussed, in recent years a wide range of studies have investigated the use of telerehabilitation, consequently, the number of studies investigating the measurement properties of these measurement tools has also increased [17–19, 22]. Systematic reviews are considered the best way to synthesize existing information and provide a comprehensive analysis of the full range of literature on a particular topic [23]. To our knowledge, no systematic review gathered information on the measurement properties and clinical utility of the measurement tools used in telerehabilitation in individuals with neurological conditions. Given the growth

of telerehabilitation in research and clinical practice [12, 16] and the challenges of using measurement tools in remote evaluation [20], a systematic review is necessary. Furthermore, this is an important source of information for researchers and professionals to choose measurement tools with adequate measurement properties and clinical utility. Therefore, this systematic review aims to investigate the measurement properties and the clinical utility of measurement tools used in telerehabilitation in individuals with neurological diseases.

## Methods

This systematic review protocol was conducted following Preferred Reporting Items for Systematic Review and Meta-Analysis Protocols (PRISMA-P) [24, 25] and the systematic review results will be reported according to Preferred Reporting Items for Systematic Review and Meta-Analysis (PRISMA 2020 statement) [26].

### Study registration

According to PRISMA-P guidelines [24, 25], this systematic review protocol was registered in the International Prospective Register of Systematic Reviews (PROSPERO) on 28 May 2021 (registration number: CRD42021257662).

### Eligibility criteria

All full-text papers that aimed to investigate the measurement properties of measurement tools used in telerehabilitation in adults (age ≥18 years old), who had a neurological condition, for example, stroke, Parkinson's disease, spinal cord injuries, multiple sclerosis, cerebellar ataxia, traumatic brain injuries, and peripheral nervous system diseases, will be included. The searches will not be limited by study design, language, or date of publication.

Research reports, working papers, conference proceedings, conference abstracts, commentaries, letters, dissertations, theses, and editorial papers will be excluded. Systematic reviews and qualitative studies will also be excluded, but their reference lists will be screened for relevant studies.

### Search strategy for identification of relevant studies

Electronic search will be carried out for articles indexed on following databases: Medical Literature Analysis and Retrieval System Online (MEDLINE Ovid), Excerpta Medica Database (Embase Classic + Embase Ovid), Physiotherapy Evidence Database (PEDro), Scientific Electronic Library Online (Scielo), and Literatura Latino-Americana e do Caribe em Ciências da Saúde (LILACS). The search strategy was designed according to previous studies and with the assistance of an experienced researcher. The established search strategy for the MEDLINE database will be adapted to suit the other databases. The search strategy (S1 File) is composed of blocks of key terms related to the target population, telerehabilitation, as measurement properties, as follows:

Target population: Individuals with neurological conditions. The search was developed based on a systematic review by Marinho-Buzelli et al. 2015 [27].

Telerehabilitation: The search was developed based on a Cochrane systematic review by Laver et al. 2020 [28].

Measurement properties: The search was developed based on a systematic review by Silva et al. 2014 [29].

## Selection of the studies

The selection of the studies will be carried out in three stages. In the first stage, searches will be carried out in the databases. Then, it will be saved and maintained in the Rayyan Systems Inc software [30]. In the second stage, the reviewers will screen the titles and abstracts of the records for eligibility, and the duplicates will be removed. In the third stage, selected full-text articles will be screened for eligibility. The excluded studies and the reason for the exclusion will be recorded. The selection will be performed by two reviewers (SAFB and PCP) independently. Disagreements between reviewers will be resolved by consensus or by the decision of a third independent reviewer (CDCMF). During the selection and screening of the studies, reviewers will be blinded to authors, journals, and outcomes. If additional information would be necessary, the authors of the paper will be contacted. According to the PRISMA 2020 statement [26], a study flow diagram will be created to depict the flow of information through the different stages of this systematic review.

## Data extraction

Two reviewers (SAFB and PCP) will independently extract data from the articles using a pre-designed data extraction form. The relevant data extracted from all the included studies will be summarized in tables. Disagreements between reviewers will be resolved by consensus or by the decision of a third independent reviewer (CDCMF).

The tables will contain the following data: study authors; year and country of publication; participants and settings (including data on age, gender, type of neurological condition, and severity of the condition); name, outcomes, and characteristics of the measurement tools; settings and mode of administration (such as telephone, videoconferencing, and sensors); measurement properties (internal consistency, reliability, measurement error, content validity, face validity, construct validity, structural validity, cross-cultural validity, criterion validity, responsiveness); clinical utility; and methodological quality of the study.

## Assessment of the methodological quality of the included studies

The methodological quality of the included studies will be assessed using the Consensus-based Standards for the selection of health Measurement Instruments (COSMIN) Risk of Bias checklist [31, 32]. In the COSMIN checklist three domains are distinguished (reliability, validity, and responsiveness), and each domain contains one or more measuring properties. The COSMIN checklist is composed of ten boxes (one box for instrument development and nine boxes for the measurement properties). The boxes contain various items, and each item can be scored on a 4-point rating scale (i.e. very good, adequate, doubtful, inadequate). Standards that are considered not applicable can be skipped. An overall quality score can be obtained by taking the lowest rating for each item in one box ("worst score counts" method) for each measurement property [31–33]. Following the recommendation by Prinsen et al. 2018 [34], the criteria for good measurement properties will also be used [34, 35]. The criteria for good measurement properties present pooled or summarized results per measurement property per measurement tool. Each item can be scored as sufficient (+), insufficient (−), or indeterminate (?) [34, 35]. The assessment will also be carried out by two reviewers independently (SAFB and PCP). Disagreements will be discussed between the two reviewers and, if necessary, a third reviewer (CDCMF) will be consulted.

## Assessment of the clinical utility of the identified tools

The clinical utility can be characterized by the ease with which an instrument is incorporated into clinical practice [36, 37]. This can be evaluated by the instrument's ability to be brief and

simple to apply, understand and score [23]. Parameters used to investigate clinical utility include the acceptability (individual friendliness of the instrument, often characterized by the total time to complete the tool and is influenced by the total number of items and the interpretability of the items) and the feasibility (ease of use, such as required specific training, costs, the need of supervision during the completion of the instrument, and time needed to score) [37]. In addition, the clinical utility can be assessed by criteria that may influence the clinicians in using a measurement tool in their practice, such as time to administer, analyze and interpret the measure, cost, need of specialized equipment/training, portability, and accessibility [36]. Studies that have investigated clinical utility using any of these definitions will be included.

Moreover, the clinical utility of all measurement tools will be collected and reported using the scale proposed by Tyson & Connell [36]. This is a 10-point scale that assessed the following criteria:

- Time to administer, analyze and interpret the measure (<10 minutes = 3 points, 10–30 minutes = 2 points, 30–60 minutes = 1 point, >1 hour = 0 points)

- Cost (<£100 = 3 points, £100–£500 = 2 points, £500–£1,000 = 1 point, >£1000 or unknown = 0 point)

- Need of specialized equipment/training ('No' = 2 points, 'Yes, but only simple, easy to use equipment which does not need specialist training' = 1 point, 'Yes' or 'Unknown' = 0 points)

- Portability, and accessibility ('Yes, easily (can go in pocket)' = 2 points, 'Yes, in a briefcase or trolley' = 1 point, 'No or very difficult' = 0 points) [36].

The assessment will also be carried out by two reviewers independently (SAFB and PCP).

### Data synthesis

A narrative synthesis will be carried out, which will provide texts and tables to synthesize and discuss the data of the studies and methodological characteristics, as previously described in the data extraction section. In addition, texts and tables will be used to summarize the findings regarding the methodological quality [31, 32], quality of measurement properties [33–35], and clinical utility [36, 37].

## Discussion

According to our knowledge, this systematic review will be the first one to assess the measurement properties and clinical utility of measurement tools used in telerehabilitation in individuals with neurological conditions. The purpose is to provide a discussion of the strengths and limitations of the different tools used in the evaluation performed during telerehabilitation of individuals with neurological conditions.

Remote delivery health services have been used for a few years. Recently, its use has increased in rehabilitation services, mainly in some specialties such as neurological and cardiac rehabilitation [16]. In the global crisis caused by the COVID-19 pandemic, the rehabilitation services had to adapt quickly to continue offering treatment to the patients [22]. One solution found was telerehabilitation since it allows the rehabilitation of individuals by substituting the traditional face-to-face approach, thus respecting safety rules preconized by health organizations, such as social distancing [38]. Therefore, telerehabilitation has increased exponentially in the last year, both in research and in clinical practice [22].

Investigating the measurement properties of the measurement tools is important to ensure an adequate evaluation of the outcomes [23]. The therapist-patient interaction, characteristics, and settings are different in telerehabilitation when compared to face-to-face rehabilitation

[16]. This may reflect different measurement properties compared to those previously investigated in the face-to-face evaluation. Furthermore, functional evaluation has been identified as a challenge to telerehabilitation practice [20, 39]. The identification and use of measurement tools that have appropriate measurement properties and clinical utility may enhance the feasibility and credibility of the evaluation performed remotely, and the comparability of interventions carried out by telerehabilitation. The results of this systematic review will be useful to assist physiotherapists in choosing the measurement tools they will use in telerehabilitation practice. Moreover, it can direct the definition of future research goals and the planning of new research studies.

This review employs a systematic, clear, and replicable inclusion and exclusion criteria and search strategy, as well as the approach regarding the searching, screening, and extracting data. The methods and instruments used in this study are recommended and validated. The involvement of two reviewers in all stages from the selection to the data extraction phase, as well as the assessment of the methodological quality of the included studies, will enhance the methodological rigor and credibility of the results found. The results from this systematic review will be spread by scientific peer-review publications and presentations at conferences and scientific events.

## Supporting information

**S1 Checklist. PRISMA-P (Preferred Reporting Items for Systematic review and Meta-Analysis Protocols) 2015 checklist: Recommended items to address in a systematic review protocol**[*].
(DOC)

**S1 File. Search strategy.**
(DOCX)

## Author Contributions

**Conceptualization:** Sherindan Ayessa Ferreira de Brito, Aline Alvim Scianni, Paula da Cruz Peniche, Christina Danielli Coelho de Morais Faria.

**Investigation:** Sherindan Ayessa Ferreira de Brito, Aline Alvim Scianni, Paula da Cruz Peniche, Christina Danielli Coelho de Morais Faria.

**Methodology:** Sherindan Ayessa Ferreira de Brito, Aline Alvim Scianni, Paula da Cruz Peniche, Christina Danielli Coelho de Morais Faria.

**Project administration:** Sherindan Ayessa Ferreira de Brito, Aline Alvim Scianni, Paula da Cruz Peniche, Christina Danielli Coelho de Morais Faria.

**Supervision:** Christina Danielli Coelho de Morais Faria.

**Writing – original draft:** Sherindan Ayessa Ferreira de Brito, Aline Alvim Scianni, Paula da Cruz Peniche, Christina Danielli Coelho de Morais Faria.

**Writing – review & editing:** Sherindan Ayessa Ferreira de Brito, Aline Alvim Scianni, Paula da Cruz Peniche, Christina Danielli Coelho de Morais Faria.

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
