## [Decision Letter · Decision Letter 0]

21 Dec 2021

PONE-D-21-29425Measurement properties of outcome measures used in neurological telerehabilitation: a systematic review protocolPLOS ONE

Dear Dr. Faria,

Thank you for submitting your manuscript to PLOS ONE. It has been reviewed by two experts in the field, and whilst they believe the submission to be interesting and well-written, they do not feel that it is publishable in its current form. I, therefore, invite you to submit a revised version of the manuscript that addresses the points raised during the review process.

Reviewer 2 raises a number of very helpful points about factors that could be included or clarified, and I suggest that you think about how they can best be followed. Reviewer 1 also feels that the novelty of this particular endeavour is not apparent, and so it would also be good to include a firmer sense of how this systematic review could fill some gaps that currently exist.

We look forward to receiving your revised manuscript.

Kind regards,

Alastair D. Smith

Academic Editor

PLOS ONE

Journal Requirements: 

Reviewers' comments:

Reviewer's Responses to Questions

**Comments to the Author**

1. Does the manuscript provide a valid rationale for the proposed study, with clearly identified and justified research questions?

Reviewer #1: Partly

Reviewer #2: Yes

2. Is the protocol technically sound and planned in a manner that will lead to a meaningful outcome and allow testing the stated hypotheses?

Reviewer #1: Partly

Reviewer #2: Yes

3. Is the methodology feasible and described in sufficient detail to allow the work to be replicable?

Reviewer #1: Yes

Reviewer #2: Yes

4. Have the authors described where all data underlying the findings will be made available when the study is complete?

Reviewer #1: No

Reviewer #2: Yes

5. Is the manuscript presented in an intelligible fashion and written in standard English?

Reviewer #1: Yes

Reviewer #2: Yes

6. Review Comments to the Author

You may also provide optional suggestions and comments to authors that they might find helpful in planning their study.

Reviewer #1: In this article the authors investigate the measurement properties and the clinical utility of measurement used in telerehabilitation in individual with neurological disease.

Authors should better structure the manuscript and report the novel finding if any regarding this issue.

The used of the reviewed articles should be used to answer the main questions under study.

Although this is a well written paper with a quite extensive literature survey the major problem is it is not so “innovative”, and the lack of the results strongly limit the decision to proceed with publication.

Reviewer #2: Thank you for asking me to review this systematic review protocol. Given the increasing focus on telehealth and telerehabilitation, this review is timely as it aims to investigate the measurement properties and the clinical utility of measurement instruments used in telerehabilitation in individuals with neurological diseases.

I do have some suggestions and comments:

• How will “telerehabilitation” be defined? Is there a standard definition currently in use?

• Why were these neurological conditions such as stroke, Parkinson's disease, spinal cord injuries, multiple sclerosis, cerebellar ataxia, traumatic brain injuries, and peripheral nervous system diseases selected? What about other neurological conditions or co-morbidities? Will they be excluded?

• I assume you will be including only quantitative studies. If so, you will need to state that you will exclude qualitative studies.

• What will you do with articles which are in language other than English? How will you translate those?

• Will databases such as CINALH/Emcare (allied health specific databases), Cochrane, Scopus/Web of Science be searched?

• How will you address publication and location bias? Are you planning on searching grey literature?

• Will the search terms include both keywords and MeSH?

• Extraction of data is not part of study selection section. Please remove.

• How will discrepancies in data extraction be resolved?

• I am unclear what “systematic narrative synthesis” means. Could you provide more detailed overview of this process?

• Minor point – in some instances, the tense is future (which is correct as this is a protocol) and in other instances, it is in past tense (incorrect). Please amend.

7. PLOS authors have the option to publish the peer review history of their article (what does this mean?). If published, this will include your full peer review and any attached files.

Reviewer #1: No

Reviewer #2: No

---

## [Author Response · Author response to Decision Letter 0]

14 Jan 2022

Reviewer's Responses to Questions

Comments to the Author

1. Does the manuscript provide a valid rationale for the proposed study, with clearly identified and justified research questions?

Reviewer #1: Partly

Reviewer #2: Yes

As suggested, the text has been revised to meet this requirement, as follows:

Lines 79-84: “Telerehabilitation can be defined as the remote delivery of rehabilitation services using information and communication technologies, such as telephone, videoconferencing, and sensors [12]. Telerehabilitation sessions can be used to assess, goal‐setting, intervene, education, and monitoring [2]. The use of telerehabilitation has increased over time as technologies become increasingly prevalent and easily accessible [12].”

Lines 110-115: “As previously discussed, in recent years a wide range of studies have investigated the use of telerehabilitation, consequently, the number of studies investigating the measurement properties of these measurement tools has also increased [17-19, 22]. Systematic reviews are considered the best way to synthesize existing information and provide a comprehensive analysis of the full range of literature on a particular topic [23].”

Lines 117-122: “Given the growth of telerehabilitation in research and clinical practice [12,16] and the challenges of using measurement tools in remote evaluation,20 a systematic review is necessary. Furthermore, this is an important source of information for researchers and professionals to choose measurement tools with adequate measurement properties and clinical utility.”

2. Is the protocol technically sound and planned in a manner that will lead to a meaningful outcome and allow testing the stated hypotheses?

Reviewer #1: Partly

Reviewer #2: Yes

As suggested, the text has been revised to meet this requirement, as follows:

Lines 163-169: “The selection of the studies will be carried out in three stages. In the first stage, searches will be carried out in the databases. Then, it will be saved and maintained in the Rayyan Systems Inc software [30]. In the second stage, the reviewers will screen the titles and abstracts of the records for eligibility, and the duplicates will be removed. In the third stage, selected full-text articles will be screened for eligibility. The excluded studies and the reason for the exclusion will be recorded. The selection will be performed by two reviewers (SAFB and PCP) independently.”

Lines 169-171: “Disagreements between reviewers will be resolved by consensus or by the decision of a third independent reviewer (CDCMF).”

Lines 239-243: “A narrative synthesis will be carried out, which will provide texts and tables to synthesize and discuss the data of the studies and methodological characteristics, as previously described. In addition, texts and tables will be used to summarize the findings regarding the methodological quality [30, 31], quality of measurement properties [33; 34] and clinical utility [34-36].”

3. Is the methodology feasible and described in sufficient detail to allow the work to be replicable?

Reviewer #1: Yes

Reviewer #2: Yes

We would like to thank the reviewers for the positive comments. 

4. Have the authors described where all data underlying the findings will be made available when the study is complete?

Reviewer #1: No

Reviewer #2: Yes

Due to the characteristics of the present study (protocol of a systematic review without meta-analysis), it does not involve statistics and/or data of participants. In the future, the results of the review will be published in full.

As suggested, this information was pointed out in the manuscript text, as follows:

Lines 30-31: “Data availability: All relevant data from this study will be made available upon study completion.”

Lines 153-154: “The search strategy (S1 File) is composed of blocks of key terms related to the target population, telerehabilitation…”

5. Is the manuscript presented in an intelligible fashion and written in standard English?

Reviewer #1: Yes

Reviewer #2: Yes

We would like to thank the reviewers for the positive comments.

6. Review Comments to the Author

You may also provide optional suggestions and comments to authors that they might find helpful in planning their study.

Reviewer #1: In this article the authors investigate the measurement properties and the clinical utility of measurement used in telerehabilitation in individual with neurological disease. 

Authors should better structure the manuscript and report the novel finding if any regarding this issue. The used of the reviewed articles should be used to answer the main questions under study. Although this is a well written paper with a quite extensive literature survey the major problem is it is not so “innovative”, and the lack of the results strongly limit the decision to proceed with publication.

Firstly, it is important to point out that the present manuscript is related to a systematic review protocol. Therefore, the results cannot be reported in this manuscript. The PLOS ONE Journal has published this type of manuscript, as “REGISTERED REPORT PROTOCOL”. An example of this type of publication at PLOS ONE is: 

Tomacheusk RM et al. Measurement properties of pain scoring

instruments in farm animals: A systematic review protocol using the COSMIN checklist. PLOS ONE May 14, 2021 (https://doi.org/10.1371/journal.pone.0251435). 

In the methods section, the first paragraph, we have pointed this out, as follows: 

Lines 126-129: This systematic review protocol was conducted following Preferred Reporting Items for Systematic Review and Meta-Analysis Protocols (PRISMA-P) [24, 25] and the systematic review results will be reported according to Preferred Reporting Items for Systematic Review and Meta-Analysis (PRISMA 2020 statement) [26]. 

As pointed out by the second reviewer “given the increasing focus on telehealth and telerehabilitation, this review is timely as it aims to investigate the measurement properties and the clinical utility of measurement instruments used in telerehabilitation in individuals with neurological diseases.” 

In the initial searches, it was possible to observe that the number of publications on telerehabilitation is growing, and in 2020 several systematic reviews were published on this topic. However, no systematic review has investigated the measurement properties nor the clinical utility of measurement tools used in telerehabilitation in individuals with neurological diseases. An initial electronic search was performed, resulting in more than 30 articles on measurement properties and the clinical utility of measurement tools used in telerehabilitation in individuals with neurological diseases. Therefore, the main question will be answered with the development of the systematic review. This highlights the importance of developing a systematic review on this topic. 

This review will have important results for both research and clinical practice, including a significant number of articles. The identification and use of measurement tools that have appropriate measurement properties and clinical utility may enhance the feasibility and credibility of the evaluation performed remotely, and the comparability of interventions carried out by telerehabilitation. Furthermore, the results of this systematic review will be useful to assist professionals in choosing the measurement tools they will use in telerehabilitation practice.

Information regarding the “innovative” characteristics of the present study was included, as follows:

Lines 79-84: “Telerehabilitation can be defined as the remote delivery of rehabilitation services using information and communication technologies, such as telephone, videoconferencing, and sensors [12]. Telerehabilitation sessions can be used to assess, goal‐setting, intervene, education, and monitoring [2]. The use of telerehabilitation has increased over time as technologies become increasingly prevalent and easily accessible [12].”

Lines 110-115: “As previously discussed, in recent years a wide range of studies have investigated the use of telerehabilitation, consequently, the number of studies investigating the measurement properties of these measurement tools has also increased [17-19, 22]. Systematic reviews are considered the best way to synthesize existing information and provide a comprehensive analysis of the full range of literature on a particular topic [23].”

Lines 117-122: “Given the growth of telerehabilitation in research and clinical practice [12,16] and the challenges of using measurement tools in remote evaluation,20 a systematic review is necessary. Furthermore, this is an important source of information for researchers and professionals to choose measurement tools with adequate measurement properties and clinical utility.”

Reviewer #2: Thank you for asking me to review this systematic review protocol. Given the increasing focus on telehealth and telerehabilitation, this review is timely as it aims to investigate the measurement properties and the clinical utility of measurement instruments used in telerehabilitation in individuals with neurological diseases.

I do have some suggestions and comments:

• How will “telerehabilitation” be defined? Is there a standard definition currently in use?

As suggested, this information was included in the manuscript text, as follows:

Lines 79-84: “Telerehabilitation can be defined as the remote delivery of rehabilitation services using information and communication technologies, such as telephone, videoconferencing, and sensors [12]. Telerehabilitation sessions can be used to assess, goal‐setting, intervene, education, and monitoring [2]. The use of telerehabilitation has increased over time as technologies become increasingly prevalent and easily accessible [12].”

• Why were these neurological conditions such as stroke, Parkinson's disease, spinal cord injuries, multiple sclerosis, cerebellar ataxia, traumatic brain injuries, and peripheral nervous system diseases selected? What about other neurological conditions or co-morbidities? Will they be excluded?

These neurological conditions were used as examples only. Our search strategy was defined according to previous reviews and covered broader terms, such as Nervous System Diseases and Neuromuscular disease, as well as other neurological conditions such as Alzheimer’s Disease and Myelitis. Thus, studies identified through our search strategy that comprise individuals with any neurological condition will be included.

As suggested, the text was revised, as follows:

Lines 137-139: “…who had a neurological condition, for example, stroke, Parkinson's disease, spinal cord injuries, multiple sclerosis, cerebellar ataxia, traumatic brain injuries, and peripheral nervous system diseases, will be included.”

• I assume you will be including only quantitative studies. If so, you will need to state that you will exclude qualitative studies.

As suggested, this information was included in the manuscript text, as follows:

Lines 143-144: “Systematic reviews and qualitative studies will also be excluded, but their reference lists will be screened for relevant studies.”

• What will you do with articles which are in language other than English? How will you translate those?

The review authors are proficient in English, Portuguese, French, and Spanish. These languages comprise an extensive part of the scientific literature. If there are articles in other languages, the authors of the paper will be consulted about the existence of an English version of the material, and in its absence, a translator will be consulted.

• Will databases such as CINALH/Emcare (allied health specific databases), Cochrane, Scopus/Web of Science be searched?

The databases selected for conducting the searches were selected with the assistance of an experienced researcher, and according to previous studies. In addition, the indexing profile of each database was observed according to the revision area. Electronic searches will be performed in following databases: Medical Literature Analysis and Retrieval System Online (MEDLINE Ovid), Excerpta Medica Database (Embase Classic + Embase Ovid), Physiotherapy Evidence Database (PEDro), Scientific Electronic Library Online (Scielo), and Literatura Latino-Americana e do Caribe em Ciências da Saúde (LILACS).

The databases were selected because they cover international literature in the area of biomedicine and life science topics, covering topics from different areas, such as bioengineering, public health, clinical care, animal science, nursing, and dentistry.

Systematic reviews were not included in this review, so the Cochrane database was not used.

• How will you address publication and location bias? Are you planning on searching grey literature?

One way to avoid publication and location bias is to have no language restrictions.

 “Because of the potential for bias when search strategies are limited to databases of peer-reviewed scholarly journals, the systematic review process should also include an examination of the "grey" literature.” (Portney & Watkins, 2009) Examples include research reports, working papers, conference proceedings, dissertations, and theses.

Only full-text articles published in indexed scientific journals will be included in this study. During the stage of selection of the studies, the “grey” literature will be excluded. In addition, conference abstracts, commentaries, letters, dissertations, and editorial papers will be excluded.

As suggested, this information was revised in the manuscript text, as follows:

Lines 135-136: “All full-text papers that aimed to investigate the measurement properties of measurement tools used in telerehabilitation in adults (age ≥18 years old) …”

Lines 141-142: “Research reports, working papers, conference proceedings, conference abstracts, commentaries, letters, dissertations, theses, and editorial papers will be excluded.”

• Will the search terms include both keywords and MeSH?

Our search strategy was designed according to previous studies and with the assistance of an experienced researcher, and used a combination of controlled vocabulary and text‐word terms. This is composed of blocks of key terms related to the target population, telerehabilitation, as measurement properties. Thus, the search terms include both keywords and MeSH.

• Extraction of data is not part of study selection section. Please remove.

As suggested, the text was revised, as follows:

Lines 163-169: “The selection of the studies will be carried out in three stages. In the first stage, searches will be carried out in the databases. Then, it will be saved and maintained in the Rayyan Systems Inc software [30]. In the second stage, the reviewers will screen the titles and abstracts of the records for eligibility, and the duplicates will be removed. In the third stage, selected full-text articles will be screened for eligibility. The excluded studies and the reason for the exclusion will be recorded.”

• How will discrepancies in data extraction be resolved?

Disagreements between reviewers will be resolved by consensus or by the decision of a third independent reviewer.

As suggested, this information was included in the manuscript text, as follows:

Lines 179-181: “Disagreements between reviewers will be resolved by consensus or by the decision of a third independent reviewer (CDCMF).” 

• I am unclear what “systematic narrative synthesis” means. Could you provide more detailed overview of this process?

The expression “systematic narrative synthesis” was replaced by “narrative synthesis”. “Narrative’ synthesis’ refers to an approach to the systematic review and synthesis of findings from multiple studies that rely primarily on the use of words and text to summarise and explain the findings of the synthesis (Popay et al., 2006).”

According to PRISMA 2020, “many systematic review reports include narrative summaries of the characteristics and risk of bias across all included studies.” In addition, according to the PRISMA 2009, “following the presentation and description of each included study, as discussed above, reviewers usually provide a narrative summary of the studies. Such a summary provides readers with an overview of the included studies. It may for example address the languages of the published papers, years of publication, and geographic origins of the included studies”.

Lines 239-243: “A narrative synthesis will be carried out, which will provide texts and tables to synthesize and discuss the data of the studies and methodological characteristics, according to previously described. In addition, texts and tables will be used to summarize the findings regarding the methodological quality [30, 31], quality of measurement properties [33; 34] and clinical utility [34-36].”

• Minor point – in some instances, the tense is future (which is correct as this is a protocol) and in other instances, it is in past tense (incorrect). Please amend.

We are sorry for this mistake. As suggested, the errors have been fixed. The excerpt that remained in the past tense refers to the elaboration of the search strategy attached to the protocol. This has already been done.

Lines 151-153: “The established search strategy for the MEDLINE database will be adapted to suit the other databases.”

Lines 153-154: “The search strategy (S1 File) is composed of blocks of key terms related to the target population, telerehabilitation, as measurement properties, as follows:”

---

## [Decision Letter · Decision Letter 1]

9 Mar 2022

Measurement properties of outcome measures used in neurological telerehabilitation: a systematic review protocol

PONE-D-21-29425R1

Dear Dr. Faria,

Thank you for submitting your revised manuscript - the original reviewers have both kindly assessed the draft and we are all satisfied that the comments have been appropriately addressed. We are, therefore, pleased to inform you that your manuscript has been judged scientifically suitable for publication and will be formally accepted for publication once it meets all outstanding technical requirements.

Kind regards,

Alastair D. Smith

Academic Editor

PLOS ONE

Additional Editor Comments (optional):

Reviewers' comments:

Reviewer's Responses to Questions

**Comments to the Author**

1. Does the manuscript provide a valid rationale for the proposed study, with clearly identified and justified research questions?

Reviewer #1: Yes

Reviewer #2: Yes

2. Is the protocol technically sound and planned in a manner that will lead to a meaningful outcome and allow testing the stated hypotheses?

Reviewer #1: Yes

Reviewer #2: Yes

3. Is the methodology feasible and described in sufficient detail to allow the work to be replicable?

Reviewer #1: Yes

Reviewer #2: Yes

4. Have the authors described where all data underlying the findings will be made available when the study is complete?

Reviewer #1: Yes

Reviewer #2: Yes

5. Is the manuscript presented in an intelligible fashion and written in standard English?

Reviewer #1: Yes

Reviewer #2: Yes

6. Review Comments to the Author

You may also provide optional suggestions and comments to authors that they might find helpful in planning their study.

Reviewer #1: No additional comments ,,,,,,,,,,,,,,,,,,,,,,,,,,,,,,,,,,,,,,,,,,,,,,,,,,,,,,,,,,,,,,,,,,,,,,,,,,,,,,,

Reviewer #2: Thank you for addressing my comments and suggestions. I am happy with the revisions that have been undertaken.

7. PLOS authors have the option to publish the peer review history of their article (what does this mean?). If published, this will include your full peer review and any attached files.

Reviewer #1: No

Reviewer #2: No

---

## [Editor Report · Acceptance letter]

11 Mar 2022

PONE-D-21-29425R1 

Measurement properties of outcome measures used in neurological telerehabilitation: a systematic review protocol 

Dear Dr. Faria:

I'm pleased to inform you that your manuscript has been deemed suitable for publication in PLOS ONE. Congratulations! Your manuscript is now with our production department. 

Kind regards, 

on behalf of

Dr Alastair Smith 

Academic Editor

PLOS ONE